# The effects of weather and mobility on respiratory viruses dynamics before and during the COVID-19 pandemic in the USA and Canada

Irma Varela-Lasheras[1], Lilia Perfeito[2], Sara Mesquita[2,3], Joana Gonçalves-Sá[1,2]*

**1** Nova School of Business and Economics, Universidade Nova de Lisboa, Carcavelos, Portugal, **2** LIP, Laboratório de Instrumentação e Física Experimental de Partículas, Lisbon, Portugal, **3** Nova Medical School, Universidade Nova de Lisboa, Lisbon, Portugal

* joanagsa@lip.pt

## Abstract

The flu season is caused by a combination of different pathogens, including influenza viruses (IVS), that cause the flu, and non-influenza respiratory viruses (NIRVs), that cause common colds or influenza-like illness. These viruses exhibit similar dynamics and meteorological conditions have historically been regarded as a principal modulator of their epidemiology, with outbreaks in the winter and almost no circulation during the summer, in temperate regions. However, after the emergence of SARS-CoV2, in late 2019, the dynamics of these respiratory viruses were strongly perturbed worldwide: some infections displayed near-eradication, while others experienced temporal shifts or occurred "off-season". This disruption raised questions regarding the dominant role of weather while also providing an unique opportunity to investigate the roles of different determinants on the epidemiological dynamics of IVs and NIRVs. Here, we employ statistical analysis and modelling to test the effects of weather and mobility in viral dynamics, before and during the COVID-19 pandemic. Leveraging epidemiological surveillance data on several respiratory viruses, from Canada and the USA, from 2016 to 2023, we found that whereas in the pre-COVID-19 pandemic period, weather had a strong effect, in the pandemic period the effect of weather was strongly reduced and mobility played a more relevant role. These results, together with previous studies, indicate that behavioral changes resulting from the non-pharmacological interventions implemented to control SARS-CoV2, interfered with the dynamics of other respiratory viruses, and that the past dynamical equilibrium was disturbed, and perhaps permanently altered, by the COVID-19 pandemic.

## Author summary

The flu season, which results in millions of cases of severe illness and hundreds of thousands of deaths worldwide, per year, is caused not only by influenza viruses but also by other respiratory viruses that cause similar symptoms. These viruses have similar

**Data Availability Statement:** This work used exclusively public data. The incidence was obtained from The National Respiratory and Enteric Virus Surveillance System (NREVSS) (available from:

https://www.cdc.gov/surveillance/nrevss/),
FluView, U.S. Influenza Surveillance. Influenza
Division, CDC (available from: https://www.cdc.
gov/flu/weekly/overview.htm), the Centre for
Immunization and Respiratory Infectious Diseases,
Public Health Agency of Canada (available from:
https://www.canada.ca/en/public-health/services/
surveillance/respiratory-virus-detections-canada.
html), and the Infectious Diseases, Public Health
Agency of Canada (available from: https://www.
canada.ca/en/public-health/services/diseases/flu-
influenza/influenza-surveillance/weekly-influenza-
reports.html) The weather data was obtained from
the Iowa Environmental Mesonet, Iowa State
University (available from: https://mesonet.agron.
iastate.edu/request/download.phtml) The mobility
data was obtained from COVID-19 Related
Transportation Statistics, Bureau of Transportation
Statistics. US Department of Transportation
(available from: https://www.bts.gov/covid-19),
and the Google COVID-19 Community Mobility
Reports (available from: https://www.google.com/
covid19/mobility/) For increased transparency and
to facilitate reproducibility of our work all datasets
and code used to analyse them are freely available
at Zenodo (https://doi.org/10.5281/zenodo.
10000040 and https://doi.org/10.5281/zenodo.
10135395).

**Funding:** This work was partially funded by FCT
grant DSAIPA/AI/0087/2018 to JGS and by PhD
fellowship 2020.10157.BD to SM. The funders had
no role in study design, data collection and
analysis, decision to publish or preparation of the
manuscript.

**Competing interests:** The authors have declared
that no competing interests exist.

circulation patterns, with outbreaks in the winter and almost no activity in the summer.
Weather has been considered a main driver of their dynamics but, after the start of the
COVID-19 pandemic, these dynamics changed so drastically that questions were raised
regarding the relative roles of different factors. We used data on several respiratory
viruses, from Canada and the USA, and tested the effects of weather and mobility in their
dynamics before and during the COVID-19 pandemic. Using statistical modelling, we
found that, whereas in the pre-COVID-19 pandemic period weather had a strong effect
and mobility a limited effect, in the pandemic period the effect of weather was strongly
reduced while mobility played a more relevant role. These results might help us better
understand the complex system of interactions between different factors that drive respi-
ratory virus dynamics and have important consequences for public health policies.

## Introduction

The flu season causes about 3 to 5 million cases of severe illness and hundreds of thousands of
deaths worldwide per year [1], posing a strong socioeconomic burden on societies and public
health systems [2]. Despite being called "flu season", influenza viruses (IVA and IVB) circulate
in parallel with other non-influenza respiratory viruses (NIRVs), which frequently account for
more than half of the influenza-like illness (ILI) cases in a season [3, 4]. Furthermore, NIRVs
significantly contribute to the morbidity and mortality of this seasonal epidemic. For example,
respiratory syncytial virus (RSV) is the predominant viral pathogen associated with acute
lower respiratory infection in children younger than 5 [5]; common human coronaviruses
(hCoVs) are responsible for 15–30% of upper respiratory track infections and can cause more
severe disease in vulnerable groups (neonates, children, elderly, and in individuals with
comorbidities) [6]; and metapneumovirus (hMPV) is a leading cause of acute respiratory
infection particularly in those same groups [7, 8].

As mentioned above, in temperate regions, IVs and NIRVs co-circulate and have similar
seasonality patterns, typically peaking in the winter and almost disappearing during the sum-
mer [9]. In fact, their dynamics are often mimicked using different types of time series analyses
and variations of compartmental models, such as the Susceptible-Infected-Recovered (SIR),
with cyclic variation assured by exhausting/replenishing the number of susceptible individuals,
by fragmenting populations, or by adding external cyclic parameters, such as weather [10–12].
Indeed, the effect of climate in viral circulation, particularly temperature and humidity, has
been widely accepted as a main factor responsible for the observed seasonal dynamics in both
hemispheres: first, there is biological evidence showing that weather conditions can influence
virus survival [13–17], transmission efficiency [15, 16, 18], and host susceptibility [19, 20]; sec-
ond, there is epidemiological evidence showing that weather conditions play an important role
in IV epidemics [21–24], although there is an ongoing discussion regarding their relative roles
(see [15, 16]); and third, adding weather variables often improves predictive models [21, 25–
30]. However, the COVID-19 pandemic and the measures implemented to contain its spread
have deeply perturbed IVs' and NIRVs' dynamics: not only the circulation of IVs and NIRVs
was reduced to a great extent, which could be explained by the great reduction in physical con-
tact but, in some cases, and quite unexpectedly, their seasonal patterns were also altered [31–
34], with local epidemics appearing "off-season".

Weather was never presented as the only responsible for seasonal epidemics, and other
commonly identified drivers of seasonal patterns include viral interactions and human behav-
iour [35, 36]. Viral interactions, either positive (synergistic) or negative (antagonistic), have

been shown to occur between respiratory viruses, with potential implications for their epidemiological dynamics. Positive interactions seem less common [37], but evidence for negative interactions exist (between IVs, between IVA, RSV and rhinovirus, between RSV and hMPV), both from epidemiological and from experimental studies, in patients and in animal and cellular models [38].

Regarding behaviour, it can impact infection numbers in at least three different ways: it may influence transmission rates through the frequency and type of social contacts (e.g. holidays, school periods, international traveling, etc) [39–43], through health-related behaviours (e.g., masking, isolation, hygiene practices, etc. [44, 45]), or at the susceptibility level, through vaccination acceptance or refusal [46].

Understanding the contribution of these different factors to the dynamics of respiratory viruses becomes even more relevant as they are varying at a global scale with climate change, newly emerging viruses, and changes in relevant human behaviours (e.g., fast urbanization, large-scale travelling or mask adoption).

However, as these drivers do not exist in isolation (different population structures, weather, viral interactions, and behaviour can influence each other) and often co-occur, disentangling between their relative contributions is a complex problem. In particular, there is a strong link between weather and human behaviour, with the first strongly influencing the second. For example, outdoor weather conditions influence the amount of time spent indoors [47] and, therefore, can potentially have an effect on both viral viability and social contact rates [43, 48].

Here, we argue that the COVID-19 pandemic and the subsequent disruption in behavioural patterns, offer an unique opportunity to test the relative importance of these different driving factors to the epidemiological dynamics of IVs and NIRVs.

We took advantage of the data publicly available from epidemiological surveillance systems on respiratory viruses, in the USA and in Canada, and tested the effects of weather and of mobility (as a relevant component of behaviour [49–54]), in viral dynamics before and after the onset of the COVID-19 pandemic. By comparing their contributions over time, we were able to better analyse their respective weights in the pre-pandemic and pandemic periods.

## Materials and methods

### Data sources, definitions, and data processing

The main criteria for data collection was availability. We 1) selected USA and Canada as these countries collect respiratory viral data throughout the year (and not only during "influenza-season") and continued to do so using similar surveillance systems during the COVID-19 pandemic, minimizing measurement bias; 2) included all respiratory viruses that were collected by both countries, for the maximum period of time available, and 3) included pandemic mobility data for both USA and Canada and also pre-pandemic mobility in the case of the USA (unavailable for Canada).

**Epidemiological data.** *Viral positivity rates* (number of positive tests for a given virus divided by the total number of tests for that virus) were collected for the USA and Canada at nation-wide levels. The two countries collect data differently and harmonization was required to allow comparison as detailed below. For the USA, human coronaviruses (hCOVs), respiratory syncytial virus (RSV) and human metapneumovirus (hMPV) data was collected from The National Respiratory and Enteric Virus Surveillance System (NREVSS, CDC), from July 2016 until January 2023. The human coronavirus positivity rates are reported for the 4 variants separately (i.e. CoVHKU1, CoVNL63, CoVOC43 and CoV229E), so assuming the tests are done in parallel (through multiplexing) and that there are no or few co-infections, these values were

summed to obtain the positivity rate for hCOVs. For IVs, the data source was FluView (CDC) [55], [56] from October 2015 until January 2023.

For Canada, data was collected from The Respiratory Virus Detection Surveillance System (FluWatch, CIRID) for the same viruses [57] from August 2013 until January 2023.

*ILI outpatient rates* (number of outpatients visits to sentinel physicians due to ILI divided by the total number of visits) were collected using weekly, nation-wide ILI data from the U.S. Outpatient Influenza-like Illness Surveillance Network (ILINet, CDC) in the case of USA [58] (from October 2015 until January 2023), and from the Syndromic/Influenza-like Illness Surveillance (FluWatch, CIRID) in the case of Canada [59] (only available from September 2016).

*Incidence* was defined as the product of the weekly virus-specific positivity rate and the weekly ILI-outpatient rate, per 1000 people, and this rate is referred to by *incidence*, throughout the text. This proxy is regarded as the best relative measure of incidence that can be calculated from surveillance data, even though it involves a series of assumptions which are often not completely met (see [60] for details). Therefore, for Canada and the USA, and for all viruses, incidence was estimated following Eq (1):

$$I(t) = p(t) \times ILI\ outpatient\ rate(t) \tag{1}$$

where $I(t)$ is incidence at time $t$, $p(t)$ is the fraction of tests that were positive for each virus at time $t$ and the ILI outpatient rates are per 1000 visits.

For Canada, it was also possible to calculate incidence by dividing the total number of positive tests for each virus by the average Canadian population between 2017 and 2023. Both incidence proxies (using either ILI outpatient rates or the average Canadian population) showed correlation coefficients ranging from 0.74 to 0.92 (S1 Fig).

**Weather data.** The weather variables Temperature (T) and Relative Humidity (RH) were selected for 1) being previously shown to affect virus transmission and survival [21, 23, 24, 61] and 2) being easy to obtain from different locations in a standardized way. Intra-daily data on temperature (˚C) and relative humidity (RH, %) were collected from Iowa Environmental Mesonet [62], from October 2015 to January 2023, for all stations available in each region.

Absolute humidity (AH, $gr/m^3$), which has also been shown to affect virus transmission and survival [15, 23, 24], was estimated following Eq (2):

$$Absolute\ Humidity(gr/m^3) = \frac{(6.1078 \times e^{17.27 \times T/(T+237.3)}) \times RH \times 2.1674}{(273.15 + T)} \tag{2}$$

where T is temperature in Celsius and RH is relative humidity in %.

To extrapolate from station data to country data, daily values per station were first averaged to obtain the weekly average (of daily averages). These were calculated first at the state or province level (for the USA and Canada, respectively) and these weekly averages per state/province were weighted according to population to give the whole country average (S2 and S3 Figs).

**Mobility data.** Mobility data includes two data sources. The first was collected from the Bureau of Transportation Statistics (US Department of Transportation, [63]) from January 2019 to March 2022 for the USA. These travel statistics are produced from an anonymized national panel of mobile device data from multiple sources and a weighting procedure expands the sample, so that the results are representative of the entire population. As these include a number of metrics, Principal Components Analysis was used to select relevant mobility variables, using as criteria low correlation with other variables, and how much variability it explained, i.e., how representative of overall mobility each metric is. The variables "Number of residents staying at home" (i.e., persons who make no trips of more than one mile away from home) and 'Number of trips made by residents" (i.e., movements that include a stay of longer

than 10 minutes at an anonymized location away from home) fulfilled both criteria and were selected for further analysis (S4 Fig). They are referred to by "Population at home" and "Number of trips", respectively (S2 Fig).

The second data source was Google's COVID-19 Community Mobility Reports (GCMR, [64]) available from February 2020 to October 2022 for both countries. This data set shows how visitors to (or time spent in) categorized places changed compared to a certain baseline day (the median value from the 5-week period between January 3 and February 6, 2020), for an aggregate of Android phone users. From this data set, the variables "Residential" and "Transit stations" were chosen for the analysis, as these show the change in total visitors to these places, and are similar in nature and correlated with the Bureau of Transportation Statistics data set described above(S2 Fig). They are referred to by "Residential time" and "Transit stations visitors", respectively (S3 Fig).

**Periods.**    Periods were defined similarly for both countries. For the weather analysis, the pre-COVID-19 pandemic period, referred to throughout the text as "pre-COVID-19", starts in October 2015 (for IVs) or July 2016 (for NIRVs) for the USA, and September 2016 for Canada (set by data availability) and ends on the week of $07^{th}$ March, 2020. The pandemic period, referred to throughout the text as "pandemic", starts on the following week, of the $14^{th}$ March, 2020, to include the date when the COVID-19 pandemic was declared by the WHO (see Fig 1), and ends on January 2023, for both countries. As a control, we have tested a pre-COVID-19 period of 2 years (from September 2016 to September 2018), similar to the pandemic period duration (see Results section).

For the mobility analysis with the US Department of Transportation data, the pre-COVID-19 period starts in January 2019 and ends on the week of $07^{th}$ March, 2020, whereas the pandemic period starts on the following week, of the $14^{th}$ March, 2020, and ends in March 2022. As a control, we have tested including the period between the week of the $18^{th}$ January 2020 to March 2020 in the"pandemic" period as there was already SARS-CoV2 circulation (see [65]). Furthermore, as these periods include different time-spans and dynamics, and do not represent unique epidemiological seasons, we have further repeated the analysis using similar periods of 15 months for the pandemic period, from January 2021 to March 2022.

Finally, for the mobility analysis with the COVID-19 Community Mobility Reports the pandemic period starts on the week of the $14^{th}$ March, 2020, and ends in March 2022.

## Data analysis

**Clustering.**    Clustering was used to identify groups of viruses that have more similar temporal dynamics. Virus-specific weekly incidence time series were normalized to the maximum of each season (starting in the last week of August), therefore considering only the variability due to timing and patterns of the waves and not their amplitude. These incidence time series were clustered using hierarchical clustering. The pairwise Euclidean distance between time series was computed and Ward's linkage method (an agglomerative algorithm) was used to construct the dendrograms. Clustering was performed in Python3, using scipy.cluster.hierarchy.dendrogram.

**Correlations.**    Correlations were used to further explore the similarities between the temporal dynamics of the viruses, as well as with the weather variables. Pairwise Pearson (and Spearman) correlations were calculated between the time series of the virus-specific weekly incidence, mean temperature, RH, and AH, after normalization to the maximum of each season, as previously described. For Pearson, the analysis was repeated including one and two week lags for the weather variables, to account for delays in notification. Correlation matrices are depicted using heatmaps with the viruses sorted by correlation to IVA. The correlation

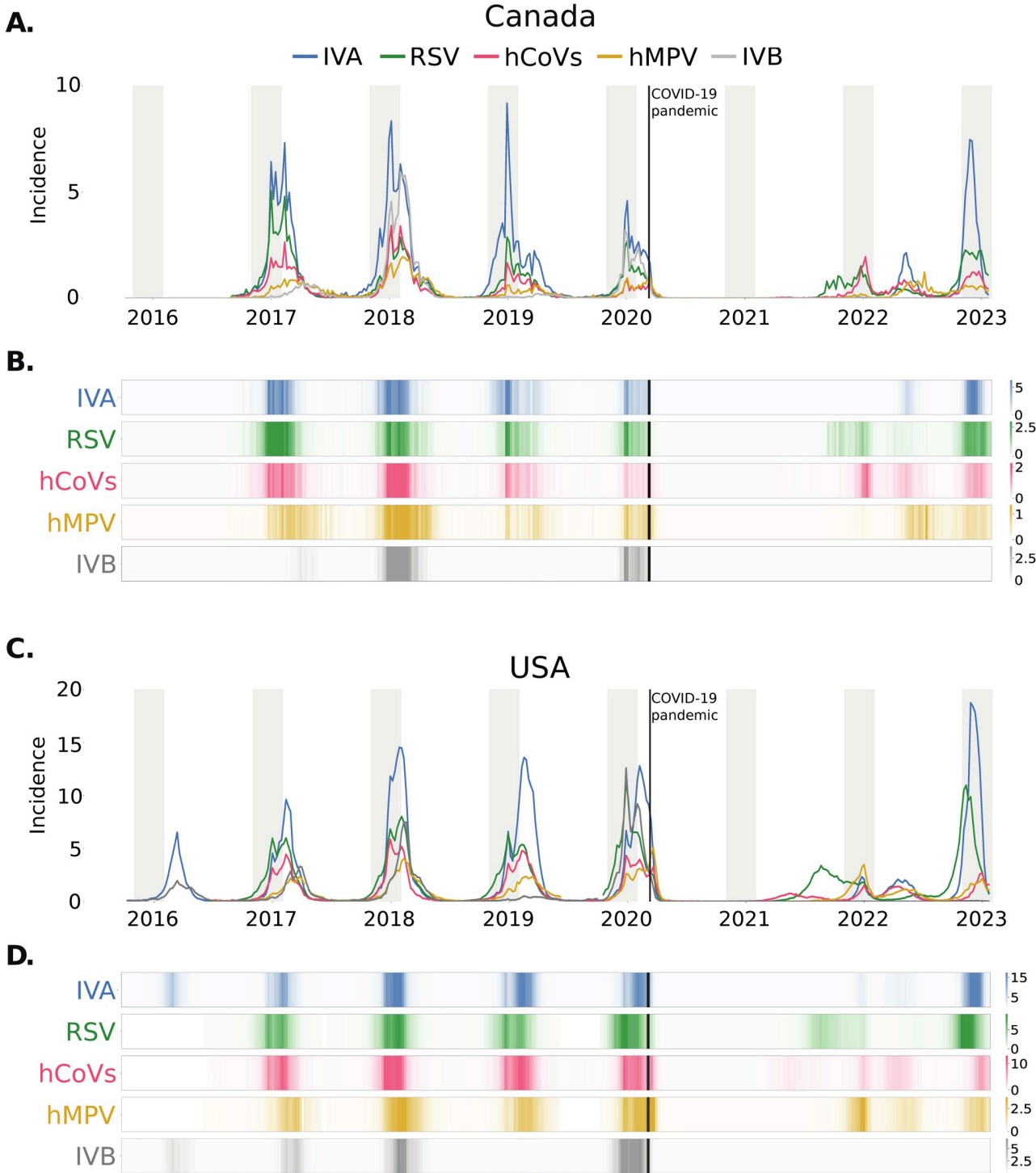

**Fig 1. Incidence per 1000 people of different respiratory viruses.** Influenza A (IVA—blue), influenza B (IVB—grey), common human coronaviruses (hCoVs—pink), human metapneumovirus (hMPV—yellow) and respiratory syncitial virus (RSV—green). Incidence in Canada **(A)** and the USA **(C)**. Shaded area corresponds to the period between November and February of the following year; solid vertical line marks the WHO pandemic declaration in March 11th, 2020, used as the separation between pre- and COVID-19 pandemic periods. Seasonal patterns for respiratory viruses and their co-occurrence in Canada **(B)** and USA **(D)**. Y axes have different scales for the different viruses and countries.

coefficients and p-values, were calculated in Python 3 using scipy.stats.pearsonr (and scipy. stats.spearmanr). The heatmaps were performed using seaborn.heatmap, where coefficients are depicted in black/white when significant and light grey when their p-value ⩽0.05.

**Regressions.** Regression models were used to investigate the associations between the incidence of the different viruses, weather conditions, and mobility.

First, incidence was transformed (dividing by 1000) to represent the proportion of people infected in the population. The effects of weather and mobility on incidence were then analyzed using Beta regression, appropriate for situations where the variable of interest is continuous and restricted to the interval (0, 1) [66]. To avoid issues with values equal to zero, $2 \times 10^{-8}$ was added to the whole incidence time series. Analysis was performed using the statsmodels. betareg in Python 3.

Second, Pearson correlation and Principal Component Analysis (PCA) were used to pre-select the independent weather variables. As temperature and AH are strongly correlated and two clusters appeared (one with T and AH, and the other with RH, S4 Fig), the weather analysis models included the following 5 variable combinations: temperature, AH, temperature-RH, AH-RH and no weather variable.

Third, incidence data was tested for auto-correlation, which was expected for two reasons: a) because it is inherent to any transmission process, meaning that the number of infections at a certain time point has an effect on the number of infections in subsequent time points; and b) because ILI-cases are seasonal, with a period of approximately one year. To account for transmission auto-correlation, the incidence in the one, two, three or four previous weeks ($Y_{t-1}$, $Y_{t-2}$, $Y_{t-3}$, $Y_{t-4}$) were included as independent variables, and are referred to as auto-correlation (AC) terms (S5 Fig). As this strongly increased the number of possible models and the one-week lag consistently offered the best model, we chose the limit the analysis to the lags around it (0, 1 or 2 week lags). Therefore, in the weather analysis 15 models were tested: the 5 weather combinations with the three AC possibilities (1, 2 or no previous week incidence). In the mobility-weather analysis, the same 5 weather combinations with and without the mobility variable were tested (10 models in total), all including the $Y_{t-1}$ AC term. Seasonal auto-correlation was evaluated in the residuals of the models, using the Breusch–Godfrey test (.acorr_-breusch_godfrey, in statsmodels). The lag parameter was 53 (weeks), to include the effects up to 1 year, except in the case of the pre-COVID-19 pandemic mobility analysis where the lag parameter was 5 weeks, given the shorter period analyzed)(S1 to S6 Tables). The null hypothesis in this test is that there is no serial correlation of any order up to the designated lag, so p-values smaller than 0.05 indicate the presence of auto-correlation in the model residuals.

All independent variables were standarized using sklearn StandardScaler in Python3.

Finally, the different models for each virus and period were compared using both $R^2$ and the Akaike Information Criterion (AIC) and considered to be identical if the difference in AIC (ΔAIC) was <= 3, and similar when the difference in AIC was <= 10. For the pseudo-$R^2$, rsquared (Python3 statsmodels) was used to compute the standard Cox-Snell version based on the likelihood ratio. For the weather-only analysis, the temperature-RH-AC model was chosen to compare the results because, first, there is previous evidence of the relevance of a combination of these two factors on the epidemiological dynamics of some respiratory virus ([16, 18]), and, second, the analysis results showed that this model represented an adequate model for comparison of all viruses, regions and periods (see Results). For the models that also include mobility, the temperature-AC, mobility-AC and temperature-mobility-AC were chosen for comparison for the same reasons. In both analyses, the model with the AC term alone was used as the null model for comparison. The results of all models can be found in the supplementary tables (S1 to S6 Tables).

To control for variation in the number of trips variable in the the mobility analysis, we created a dummy variable for the pandemic period and using the whole period available (5$^{th}$ January 2019- 19$^{th}$ March 2022), we repeated the analysis including interactions.

## Results

### Weather effect pre-COVID-19 pandemic

Different respiratory viruses co-occur during the common flu season, but their incidence and dynamics are not exactly the same. Fig 1 shows the incidence for IVA, IVB, RSV, hCoVs and hMPV, for the USA and Canada, from 2015–2016 to January 2023.

To analyze the effects of weather, we performed three different analyses: first, we looked at the different viruses and identified the ones that tend to co-circulate; second, we asked whether the different groups correlate differently with the different weather variables; and third, we used statistical modelling to infer the effect of weather on the observed dynamics. To identify viruses that have similar circulation patterns, we performed hierarchical clustering and two groups of viruses emerge from this analysis for both countries: one including IVA, RSV and hCoVs; and the other including IVB and hMPV (Fig 2A and 2D).

We then performed pair-wise correlation analyses of the standardized incidence by country, for all viruses and with the different weather variables. There is a general positive correlation between all viruses, but the two groups identified previously stand out consistently in both countries (Fig 2B and 2E). As others before us [23, 24], we found strong correlations between the viruses, temperature and AH, but weaker or no correlation with RH. Moreover, the two groups of viruses identified earlier by their dynamics reflect different patterns of correlation with the weather variables: the first shows stronger negative correlations with temperature and AH, and weak positive correlations with RH (Fig 2C and 2F), whereas the second shows weaker negative correlations with temperature and AH, and weak negative or no correlation with RH (Fig 2C and 2F). The observed correlations patterns are consistent when using Spearman correlation (S7 Fig) and do not change when considering lagged weather of up to 2 weeks (S6 Fig).

To further analyse the weight of the weather variables in the incidence, we performed a Beta regression. All models and respective parameters are summarized in the supplementary tables. For comparison purposes we decided to show the model that includes auto-correlation (AC), temperature, and relative humidity (RH), as it was systematically either among the best models and/or had very similar dynamics and regression coefficients to the best models, see S1 and S2 Tables).

This analysis showed that, for every virus, in both countries, including weather variables considerably improves the models (Tables 1 and 2). Indeed, all best models (AIC difference from the absolute best model $\Delta$AIC< = 3) included at least one weather variable (plus the AC term) (S1 Table). Furthermore, this analysis confirmed the differential effect of weather, revealing a stronger dependency on temperature (or AH) for the first group of viruses compared with the second (Fig 3A and 3C, Tables 1 and 2). Regarding RH, its effect is much weaker but still significant for many of these viruses (Fig 3A and 3C, Tables 1 and 2). Indeed, this model is able to explain a considerable proportion of the variation in the incidence of these viruses (ranging from 64% up to 91%, Tables 1 and 2). These results are consistent even when considering only two seasons (S8 Fig).

### Epidemiological dynamics during the COVID-19 pandemic

All viruses in the data set showed a pronounced decline in incidence after March 2020 (Fig 1 and S9 Fig) and remained very low until the spring-summer 2021. In the spring-summer 2021,

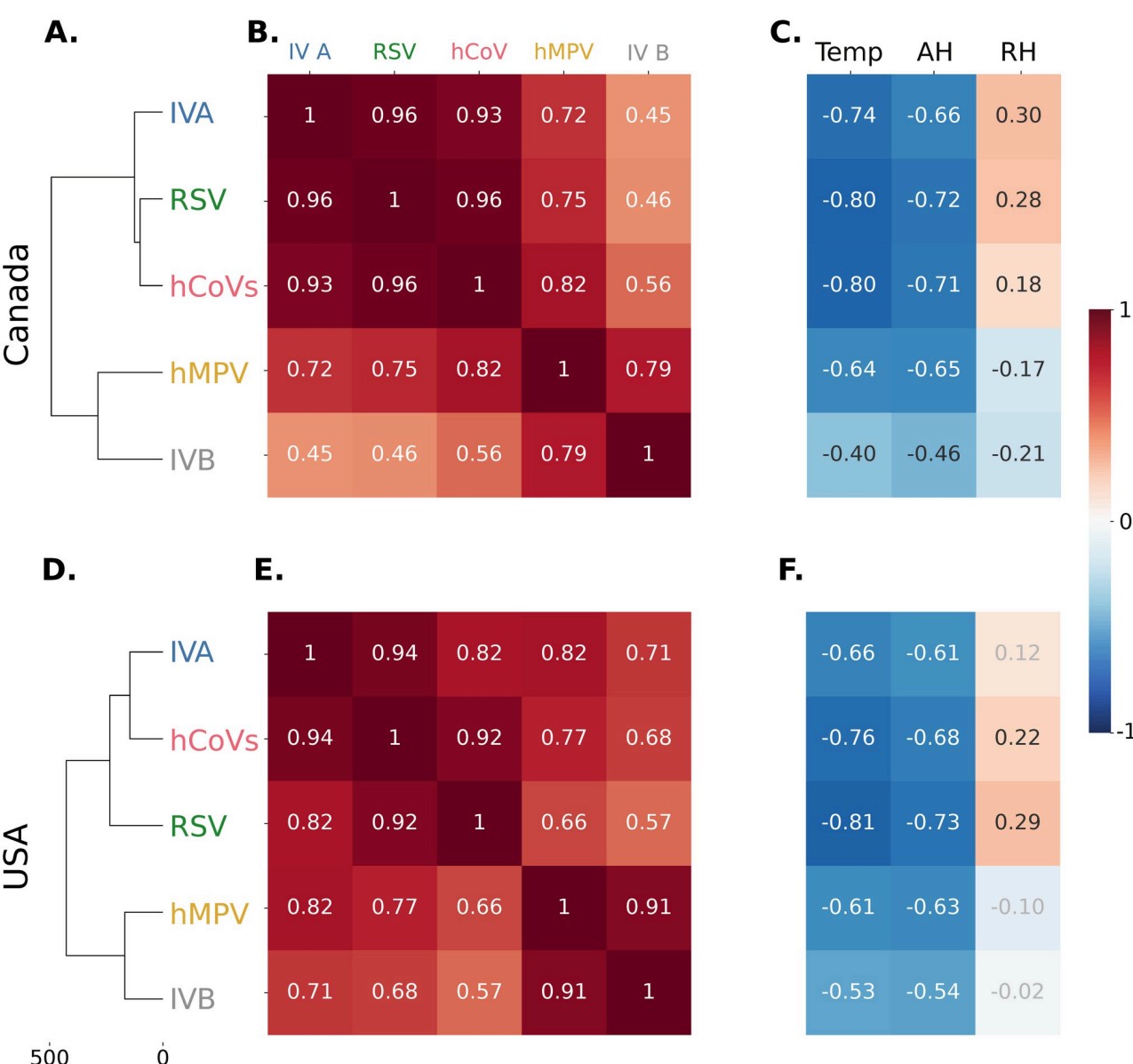

**Fig 2. Hierarchical cluster analysis and correlation patterns between different viruses and weather in the pre-COVID-19 period. (A), (D)** Hierarchical clustering dendograms for the different viruses in Canada and the USA, respectively. **(B), (E)** Correlations between the incidence of the viruses in Canada and the USA, respectively. **(C), (F)** Correlation between viral incidence and temperature, AH, and RH in Canada and the USA, respectively. Coefficients in white or black, p-value≤0.05; coefficients in light grey, non-significant.

we observed a slight increase in hCoVs activity, specially in the USA, and increased RSV activity starting in the summer 2021 until February 2022, in the USA, and just a few weeks later, in Canada. In the winter 21–22 we also observed off-season surges in IVA and hMPV in the USA, as well as in hCoVs in both countries. During the spring-summer 2022 all viruses in both countries showed increased activity or off-season surges. Similarly, an early winter epidemic during the the autumn-winter 2022–2023 was detected for all the viruses in both countries, being less intense for RSV and hMPV in Canada. Finally, IVB showed extremely low

**Table 1. Regression coefficients, ΔAIC and $R^2$ for the models that include: (1) auto-correlation (AC), (2) temperature and AC, and (3) temperature, RH, and AC for Canada pre-COVID-19 pandemic.**

|  | Model | AC | Temperature | RH | ΔAIC | $R^2$ |
|---|---|---|---|---|---|---|
| IVA | 1 | 0.67 ** |  |  | 0.0 | 0.59 |
|  | 2 | 0.37 ** | -0.77 ** |  | 139.0 | 0.81 |
|  | 3 | 0.37 ** | -0.77 ** | 0.06 ns | 139.0 | 0.81 |
| RSV | 1 | 0.61 ** |  |  | 0.0 | 0.58 |
|  | 2 | 0.35 ** | -0.78 ** |  | 155.0 | 0.82 |
|  | 3 | 0.35 ** | -0.78 ** | 0.06 ns | 155.0 | 0.82 |
| hCoVs | 1 | 0.63 ** |  |  | 0.0 | 0.59 |
|  | 2 | 0.39 ** | -0.72 ** |  | 124.0 | 0.79 |
|  | 3 | 0.39 ** | -0.74 ** | -0.11 ** | 128.0 | 0.8 |
| IVB | 1 | 0.7 ** |  |  | 0.0 | 0.54 |
|  | 2 | 0.6 ** | -0.42 ** |  | 37.0 | 0.63 |
|  | 3 | 0.59 ** | -0.46 ** | -0.16 ** | 41.0 | 0.64 |
| hMPV | 1 | 0.62 ** |  |  | 0.0 | 0.64 |
|  | 2 | 0.49 ** | -0.37 ** |  | 45.0 | 0.72 |
|  | 3 | 0.46 ** | -0.46 ** | -0.25 ** | 76.0 | 0.77 |

**, p-value ≤0.01;

*, p-value ≤0.05;

ns, non-significant.

**Table 2. Regression coefficients, ΔAIC and $R^2$ for the models that include: (1) auto-correlation (AC), (2) temperature and AC, and (3) temperature, RH, and AC for the USA pre-COVID-19 pandemic.**

|  | Model | AC | Temperature | RH | ΔAIC | $R^2$ |
|---|---|---|---|---|---|---|
| IVA | 1 | 0.86 ** |  |  | 0.0 | 0.7 |
|  | 2 | 0.64 ** | -0.59 ** |  | 103.0 | 0.81 |
|  | 3 | 0.64 ** | -0.59 ** | 0.03 ns | 102.0 | 0.81 |
| RSV | 1 | 0.76 ** |  |  | 0.0 | 0.71 |
|  | 2 | 0.46 ** | -0.77 ** |  | 182.0 | 0.9 |
|  | 3 | 0.42 ** | -0.83 ** | 0.12 ** | 207.0 | 0.91 |
| hCoVs | 1 | 0.88 ** |  |  | 0.0 | 0.76 |
|  | 2 | 0.51 ** | -0.63 ** |  | 93.0 | 0.85 |
|  | 3 | 0.51 ** | -0.66 ** | 0.13 ** | 114.0 | 0.87 |
| IVB | 1 | 0.59 ** |  |  | 0.0 | 0.55 |
|  | 2 | 0.53 ** | -0.44 ** |  | 64.0 | 0.66 |
|  | 3 | 0.56 ** | -0.47 ** | -0.21 ** | 87.0 | 0.7 |
| hMPV | 1 | 0.77 ** |  |  | 0.0 | 0.76 |
|  | 2 | 0.61 ** | -0.49 ** |  | 93.0 | 0.86 |
|  | 3 | 0.61 ** | -0.49 ** | -0.06 * | 96.0 | 0.86 |

**, p-value ≤0.01;

*, p-value ≤0.05;

ns, non-significant.

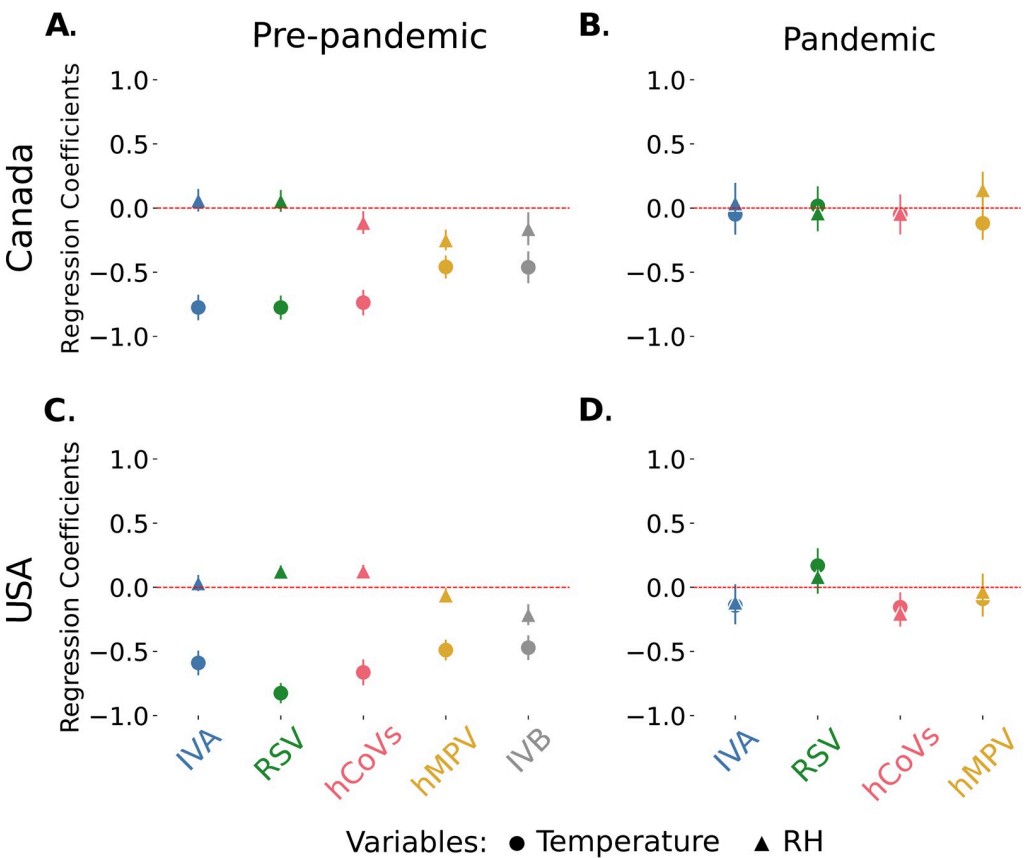

**Fig 3. Regression coefficients for the temperature-RH model in the pre-COVID-19 and pandemic periods.** Temperature and RH regression coefficients with the 95% confidence intervals for the incidence of all the viruses analyzed for **(A)** Canada pre-COVID-19 pandemic, **(B)** Canada pandemic, **(C)** USA pre-COVID-19 pandemic and **(D)** USA pandemic. Circles: temperature coefficients; Triangles: RH coefficients. Coefficients for the AC term are not represented.

levels in both countries from March 2020 (Fig 1 and S9 Fig) and, therefore, we excluded it from the pandemic period analysis.

## Weather effect during the COVID-19 pandemic

As the observed dynamics had a less obvious seasonal pattern after the WHO COVID-19 pandemic declaration, we repeated the previous analysis, on the incidence of the different viruses and their relation with weather, from March 2020 until January 2023 (Fig 1).

As expected from observing the incidence time series, and reflecting the disruption in dynamics observed in the incidence patterns for this period in Canada and the USA (Fig 1 and S9 Fig), the hierarchical clustering no longer revealed two clear groups and no clustering pattern is shared between Canada and the USA (Fig 4A and 4D). The results of the correlation analysis were consistent with the clustering results, showing much weaker correlations among the different viruses and revealing no clear groups (Fig 4B and 4E), after the pandemic started. Regarding the correlations between the viruses and weather in the same period, we observed again a much weaker correlations and no clear patterns (Fig 4C and 4F). When we repeated the analysis using Spearman correlation, we found similarly weak correlations between viruses.

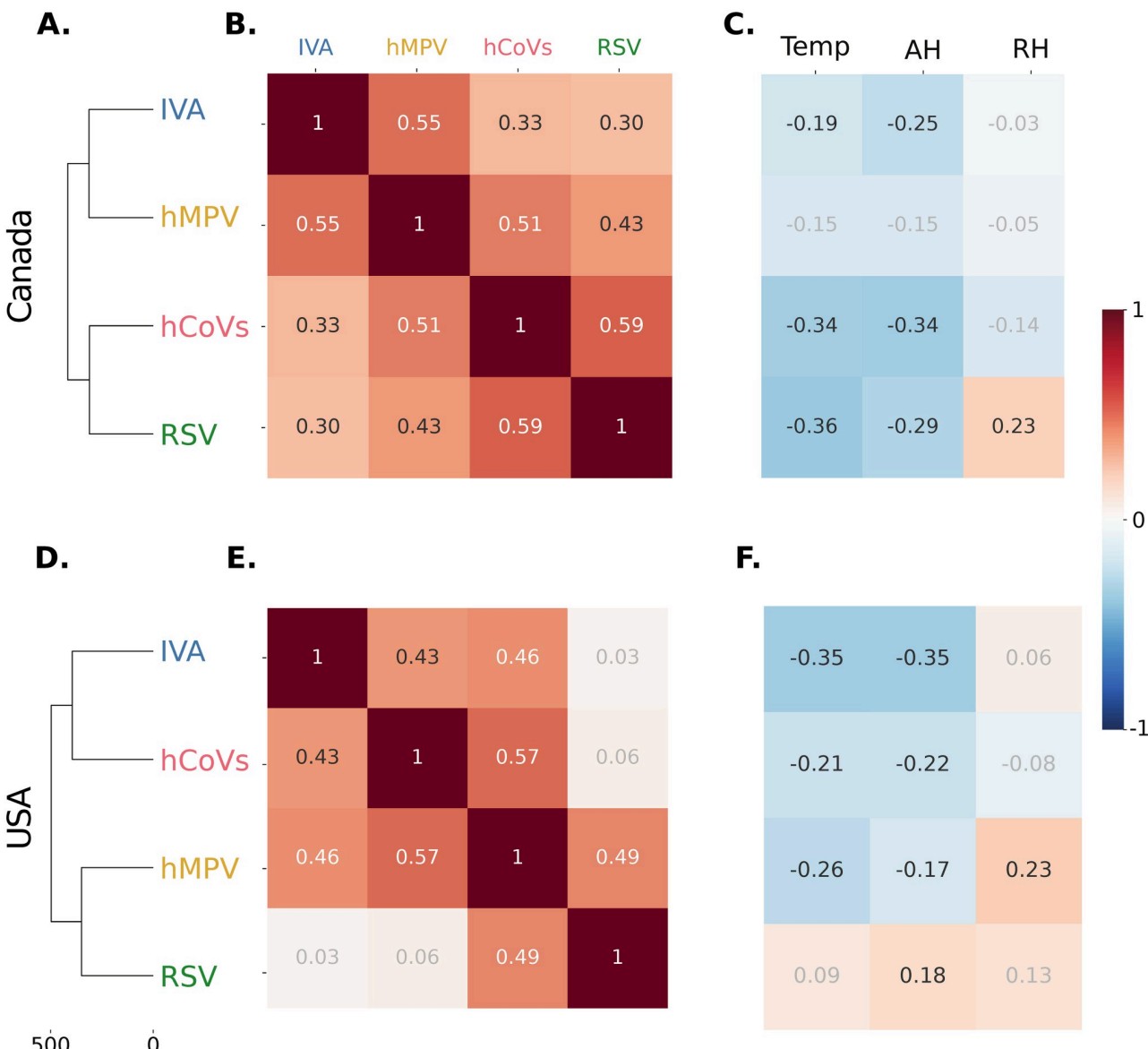

**Fig 4. Hierarchical cluster analysis and correlation patterns between different viruses and weather during the pandemic period in Canada and the USA.** (A) and (D) hierarchical clustering dendograms for the different viruses in Canada and the USA, respectively. Note that the distances between the viruses in the x-axis are larger than in the pre-COVD19 pandemic period (Fig 2). (B) and (E) Correlations between the incidence of the viruses in Canada and the USA, respectively. (C) and (F) Correlation between the incidence of the viruses and temperature, AH and RH in Canada and the USA, respectively. Coefficients in white or black, p-value≤0.05; coefficients in light grey, non-significant.

This further supports the observation that weather played a more nuanced role on viral incidence, during the pandemic period (S7 Fig).

We then performed the same regression analysis for the pandemic period. Adding a weather variable only improved the models for hMPV in Canada and, RSV and hCoVs in the USA. Moreover, it was only in the latter case that the models with and without weather were considerably different (ΔAIC>10) (S2 Table). To facilitate comparison, we again choose the model that includes temperature and RH (plus one AC term) as, again it was either among the

best models or the regression coefficients were very similar and followed similar patterns, (see S2 Table). This regression analysis confirmed a much weaker, and often non-significant effect of temperature (or AH), in the pandemic period, for all viruses and in both countries (Fig 3B and 3D and S2 Table). In fact, the models can only explain a limited proportion of the variation in viral dynamics (ranging from 40% to 65%, S2 Table), in this period. For every virus in both countries, all best models included the AC variable (S2 Table).

## Mobility effect pre- and during the COVID-19 pandemic

As mentioned before, another clear driver of epidemics is human behavior. However, given that weather models were so good at explaining seasonality of IVs and NIRVs before the COVID-19 pandemic, and that weather and behaviours are often correlated (S2 and S3 Figs), behaviour has received somewhat less attention. Therefore, that we observe not only a disruption in incidence but also in seasonality, during the pandemic period, and that these viruses are clearly able to spread during warmer months in temperate countries, raises important questions regarding the relative role of weather in their spread. Thus, we next tested whether mobility patterns, as a relevant component of behaviour, could help better understand the dynamics of the viruses in the period after the advent of COVID-19. As detailed in the methods, we could only find pre-pandemic mobility data for the USA; for Canada, only pandemic mobility is available. Therefore, we first focused in on the "number of trips" and "population at home" variables, reported by the US Department of Transportation, which allowed us to compare the effects of mobility before and during the COVID-19 pandemic in the USA. Correlation analysis show that in the pre-COVID-19 pandemic period available for this data set (January 2019-March 2020), the number of trips showed a negative correlation with the incidence of the viruses, whereas the population at home showed positive or no correlation (S2 Fig). This is possibly the result of the strong positive correlation in that single season of the number of trips and warm weather (S2 Fig). In the COVID-19 pandemic period the number of trips showed a positive correlation with the incidence of the viruses in most cases, whereas the population at home showed a negative or no correlation (S2 Fig).

Regarding the regression analysis, we compared the models with weather (plus the AC term) to models that also incorporated different mobility measures. For the pre-COVID-19 pandemic period, including the number of trips did not improve the models or the proportion of variation explained (Table 3, S3 Table). For RSV and hCoVs, the difference in AIC between the best without and with mobility models is just over 3, meaning that the model with mobility is only slightly better than the one without it (S3 Table). As seen previously for longer pre-COVID-19 pandemic periods, the effect of temperature in this single season is, in general, large and negative (Fig 5A). We also confirmed the previously observed strong positive correlation between the number of trips and weather, that makes it difficult to assess the individual effect of the mobility variable.

Similarly, including the population at home did not improve the models or the proportion of variation explained, although, again, for RSV, the model with mobility is slightly better than the one without it (S10 Fig, S4 Table).

Still using the US Department of Transportation data, we then focused on the period after the COVID-19 pandemic was declared and collected the available mobility data from March 2020 to March 2022. In contrast to the previous period, there is now no significant correlation between the number of trips and weather (S2 Fig), allowing us to better assess the individual effects of both variables. Including the total number of trips improved the models for all viruses and increased the variation explained by the models (Table 4, S3 Table), the only exception being IVA (where all models are similar, i.e., all model's AIC are within 10 units, S3

**Table 3. Regression coefficients, ΔAIC and R² for the models that include: (1) auto-correlation (AC), (2) temperature and AC, (3)number of trips and AC, and (4) temperature, number of trips and AC term for the USA pre-COVID-19 pandemic.**

|  | Model | AC | Temperature | No. Trips | ΔAIC | $R^2$ |
|---|---|---|---|---|---|---|
| IVA | 1 | 1.02 ** |  |  | 0.0 | 0.79 |
|  | 2 | 0.69 ** | -0.77 ** |  | 39.1 | 0.89 |
|  | 3 | 0.72 ** |  | -0.5 ** | 14.7 | 0.84 |
|  | 4 | 0.7 ** | -0.77 ** | 0.0 ns | 37.1 | 0.89 |
| RSV | 1 | 0.61 ** |  |  | 0.0 | 0.74 |
|  | 2 | 0.47 ** | -0.51 ** |  | 36.5 | 0.9 |
|  | 3 | 0.45 ** |  | -0.31 ** | 7.2 | 0.79 |
|  | 4 | 0.52 ** | -0.62 ** | 0.16 * | 38.0 | 0.9 |
| hCoVs | 1 | 1.04 ** |  |  | 0.0 | 0.83 |
|  | 2 | 0.65 ** | -0.7 ** |  | 37.4 | 0.91 |
|  | 3 | 0.75 ** |  | -0.35 * | 3.4 | 0.85 |
|  | 4 | 0.75 ** | -0.82 ** | 0.21 ns | 38.5 | 0.92 |
| IVB | 1 | 0.78 ** |  |  | 0.0 | 0.66 |
|  | 2 | 0.71 ** | -0.41 ** |  | 13.7 | 0.74 |
|  | 3 | 0.73 ** |  | -0.14 ns | -0.2 | 0.67 |
|  | 4 | 0.87 ** | -1.09 ** | 0.7 ** | 32.2 | 0.81 |
| hMPV | 1 | 0.55 ** |  |  | 0.0 | 0.85 |
|  | 2 | 0.5 ** | -0.14 ** |  | 6.7 | 0.88 |
|  | 3 | 0.5 ** |  | -0.09 * | 1.9 | 0.87 |
|  | 4 | 0.5 ** | -0.16 * | 0.03 ns | 4.9 | 0.88 |

**, p-value ≤0.01;

*, p-value ≤0.05;

ns, non-significant.

Note that the differences between the absolute best with and without mobility (number of trips) models for RSV and hCoVS in the pre-COVID-19 pandemic period are 3.7 and 3.3 units, respectively S4 Table.

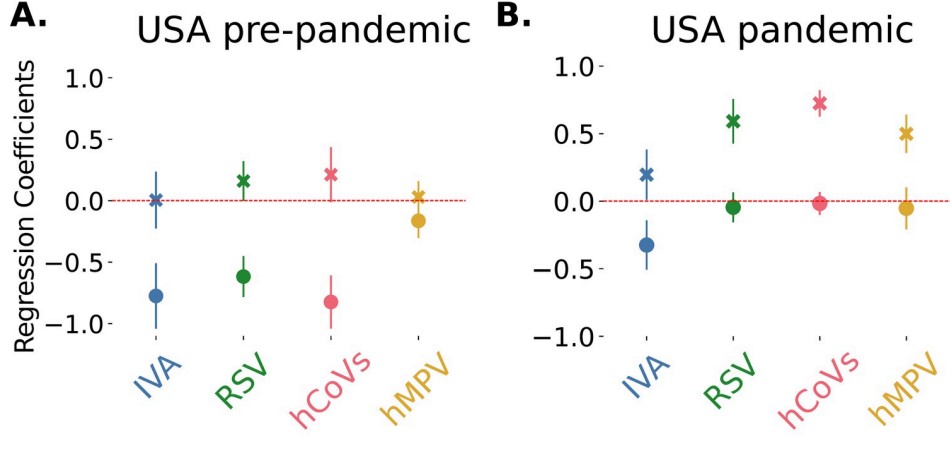

**Fig 5. Effect of number of trips in the USA.** (A) temperature and number of trips regression coefficients for the incidence of all the viruses with the 95% confidence intervals in the pre-COVID-19 pandemic period. (B) temperature and number of trips regression coefficients for the incidence of all the viruses with the 95% confidence intervals during the pandemic period.

**Table 4. Regression coefficients, ΔAIC and $R^2$ for the models that include: (1) auto-correlation (AC), (2) temperature and AC, (3)number of trips and AC, and (4) temperature, number of trips and AC term for the USA during the pandemic period.**

|  | Model | AC | Temperature | No. Trips | ΔAIC | $R^2$ |
|---|---|---|---|---|---|---|
| IVA | 1 | 0.57 ** |  |  | 0.0 | 0.42 |
|  | 2 | 0.55 ** | -0.27 ** |  | 7.0 | 0.47 |
|  | 3 | 0.57 ** |  | 0.12 ns | -1.0 | 0.43 |
|  | 4 | 0.55 ** | -0.32 ** | 0.2 * | 9.0 | 0.49 |
| RSV | 1 | 1.07 ** |  |  | 0.0 | 0.77 |
|  | 2 | 1.12 ** | -0.11 ns |  | 1.0 | 0.78 |
|  | 3 | 0.93 ** |  | 0.61 ** | 50.0 | 0.86 |
|  | 4 | 0.96 ** | -0.04 ns | 0.59 ** | 48.0 | 0.86 |
| hCoVs | 1 | 0.51 ** |  |  | 0.0 | 0.52 |
|  | 2 | 0.5 ** | -0.1 ns |  | 0.0 | 0.52 |
|  | 3 | 0.67 ** |  | 0.73 ** | 129.0 | 0.86 |
|  | 4 | 0.67 ** | -0.02 ns | 0.72 ** | 127.0 | 0.86 |
| hMPV | 1 | 0.66 ** |  |  | 0.0 | 0.85 |
|  | 2 | 0.71 ** | -0.13 ns |  | 0.0 | 0.54 |
|  | 3 | 0.88 ** |  | 0.51 ** | 41.0 | 0.68 |
|  | 4 | 0.86 ** | -0.05 ns | 0.5 ** | 39.0 | 0.68 |

**, p-value ≤0.01;

*, p-value ≤0.05;

ns, non-significant.

Table). In fact, the mobility alone model (plus the AC term) is, for every virus, among the best models(S3 Table). In these models, there is a significant positive effect of mobility in all cases, so that the fewer travels, the lower the incidence (Fig 5B, Table 4), and this effect is larger in magnitude than the effect of temperature (again, except for IVA). As the defined pandemic period does not correspond to when the virus was first identified, but only to when the pandemic was declared (March 2020), we repeated the analysis considering that the pandemic started when SARS-CoV2 was identified and known to already be in circulation (from mid January 2020). Furthermore, and given that our pre-pandemic and pandemic periods have different durations, we also considered equivalent length-periods (15 months each) to control for possible effects of differences in period. We found no significant differences in the first case (S11 Fig) and our results did not change, in the second (S12A Fig). Finally, and given the high variability present in the "number of trips" variable, during the pandemic, we also built a regression model for the whole period available, using the pandemic as a dummy variable, and found a significant interaction between the pandemic variable and the number of trips for all the virus (except IVB), supporting our previous results on the relevance of mobility for the incidence of these viruses during the pandemic (S12B Fig).

Similarly to the number of trips, including the population at home improved the models and the proportion of variation explained (except again for IVA, where all models are similar, i.e., all model's AIC are within 10 units, S10 Fig, S4 Table). Indeed, in this period, there is a significant negative effect of population at home in the incidence of most of this viruses: with the exception of IVA, the fewer people at home, the higher the incidence (S10 Fig, S4 Table), and this is true despite the defined pandemic period covering very different behaviors, from strong lock-downs to close to normal mobility (S2 and S3 Figs).

To include Canada in the analysis, we then used Google Mobility reports data, in particular the "Residential time" and the "Transit stations" categories, which allowed us to compare the effect of mobility in the available pandemic period (March 2020-October 2022) for both countries. Correlations between the incidences of the viruses and these mobility variables are in general weak, but a positive correlation was observed for visitors to transit stations whereas a negative correlation was observed for residential time (S2 and S3 Figs). This is consistent with what was observed before for the US Department of Transportation pandemic data.

Regarding the regression analysis, and as before, including the visitors at transit stations variable improved the models for all viruses, except for IVA in the USA (where all models are similar, i.e., all model's AIC are within 10 units, S5 Table). In these models, there is a significant positive effect of the mobility variable in both countries, so that the less visitors in transit stations the less incidence is found (Fig 6A and 6B). This effect is similar or larger in magnitude to the effect of temperature in all models. Similarly, including the residential time variable improved the models and the proportion of variation explained, except again for IVA in the USA (where all models are similar)(S13 Fig, S6 Table). In this period, there is a significant negative effect of residential time in the incidence of most of these viruses, so that the less time people spend in residential areas the higher the incidence (S13 Fig, S6 Table).

In summary, whereas mobility played a minimal role before the COVID-19 pandemic, it becomes fundamental to understand respiratory viral dynamics and "off-season" infections after March 2020.

## Discussion

From the different factors that impact the seasonal dynamics of respiratory viruses, weather has received the most attention [35], and there is extensive data consistent with a strong weather effect on viral dynamics. However, and given the association between weather and human behavior (e.g. school start coinciding with the end of summer or the increase in indoor activities during the colder months), understanding the relative importance of each has proven difficult.

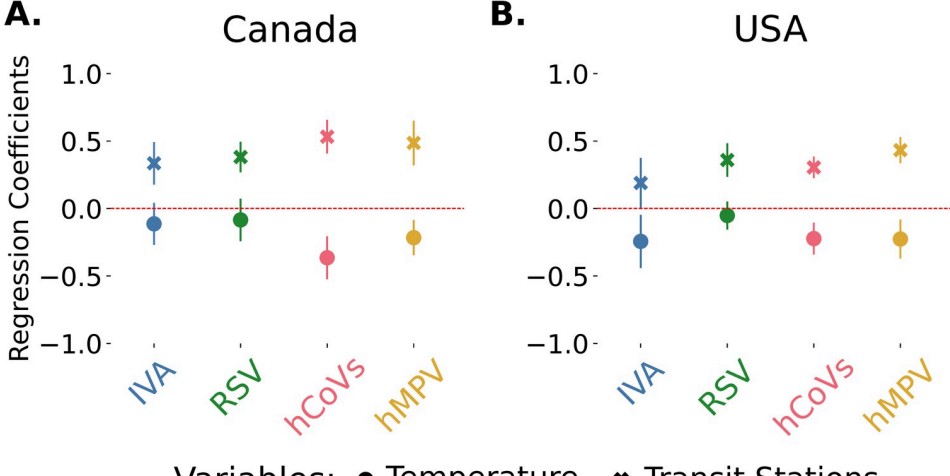

**Fig 6. Effects of visits to transit stations (GMR) in Canada and the USA.** (A) temperature and visitors to transit stations coefficients for the incidence of all the viruses with the 95% confidence intervals during the pandemic period in Canada. (B) temperature and visitors in transit stations regression coefficients for the incidence of all the viruses with the 95% confidence intervals during the pandemic period in the USA (S5 Table).

Here, we took advantage of the external shock provided by the first years of the COVID-19 pandemic and provide evidence indicating that weather had a significant effect on the epidemiological dynamics of all these viruses in the pre-COVID-19 pandemic period, but that this effect is much less significant after the pandemic started, in the USA and Canada. The effect of weather in the pre-pandemic period is significant even though this study was conducted at a national level and there is great variability in temperature, humidity and other weather variables within countries, especially in the case of the USA. Further studies could benefit from the regional level data available to investigate the specific effects of different climate types on the incidence of these viruses. This disruption in respiratory viral dynamics was observed in two different ways: first, in many temperate countries of the Northern Hemisphere, these non-SARS-CoV2 respiratory viruses virtually disappeared from March-April 2020 until at least March 2021; second, these viruses resurged but with previously unobserved dynamics, peaking outside of their traditional seasons. Several explanations have been presented for both observations and we discuss them below.

The initial reduction in IVs and NIRVs in Canada and in the USA has been reported [67, 68] and, at least for IVs and RSV, non-pharmacological interventions (NPIs), such as stay-at-home orders, travel restrictions, school closures or mask mandates, are the most commonly mentioned factor to explain it [31, 69, 70]. In the case of Canada, a significant decrease in IVs and NIRVs positivity rates was observed after the early implementation of NPIs in March 2020 [67]. Another proposed explanation is viral interference between SARS-CoV-2 and other viruses, which could have led to fewer infections but strong evidence for this effect is lacking [31, 38, 69].

The data also shows that all viruses analyzed had "off-season" surges, with peaks in the spring-summer of 2021 and/or 2022, and/or unusually early seasonal peaks in the winter of 2022–2023. These atypical dynamics have been reported for RSV and IVs around the globe [31, 70, 71] and it has been suggested that these surges may be associated with the relaxation or lifting of NPIs [31, 72]. Another leading hypothesis is immunity waning in the population due to the reduction of viral circulation in 2020 and 2021, an effect also called "immunity debt" [73, 74]. The implicit implication behind both these hypotheses is that winter weather is neither a necessary nor sufficient condition for seasonal epidemics: it may facilitate infections but, if enough individuals are susceptible and/or NPIs are not in place, summer waves could be sustained. Indeed, some countries offer relevant exceptions that contradict at least the strong temperature dependency. For example, RSV is mostly a winter virus in the Northern Hemisphere but typically peaks in the summer and fall in Japan [75]; even more interestingly, it had its highest surge on record during the spring of 2021, after having almost disappeared during the 2020 season [76]. If weather is less relevant to circulation than previously believed, some surges (in very young infants or immunocompromized groups) should be expected more frequently throughout the year. Furthermore, as summer peaks are clearly possible for some of them, global changes in weather might alter viral dynamics in ways that are difficult to predict, which is particularly relevant in the context of climate change [77, 78]. Further analysis on the effects of weather and behaviour on NIRVs is in order, both in laboratory and epidemiological settings, to inform models and allow simulation of different complex scenarios and interactions.

This effect of shifting peaks had also been observed during the 2009 flu pandemic, after a new influenza strain was identified in April 2009 (influenza A (H1N1)pdm09). This strain reached pandemic status in June but, in most of the Northern Hemisphere, only peaked in the fall. In this case, waning immunity was unlikely, as it did not lead to general and long lockdowns, and viral interference was presented as the most likely explanation: the circulation of rhinoviruses might have delayed the H1N1 wave [61], which in turn might have delayed the

RSV epidemic [79]. Still, the pandemic influenza peaked after schools resumed and the temperatures started to drop, in line with the expected roles of both mobility and weather in influenza transmission. Moreover, the subsequent epidemics, mostly driven by the pandemic strain, went back to their winter dynamics, again masking the relative roles of different drivers. In fact, mobility and mixing might be particularly relevant drivers not only behind the probability of transmission, but also in the seasonal patterns. Therefore, our results that indicate that mobility better explains the observed dynamics during COVID-19 pandemic period, are in line with previous evidence suggesting that the subsequent behavioral changes and the non-pharmacological interventions had a significant effect on the epidemiology of these viruses [80–88]. We note that this study was done at the country level and that mobility is likely to play an even more visible role when analyzing outbreaks at higher geographical granularity [89], as in USA and, to some extent, in Canada, mobility restrictions were implemented and adjusted at the state or provincial levels. Similarly, several other NPIs (including masking policies) often followed different strategies, with each territory having their own guidelines and regulations. Ideally, future studies should take regional differences into consideration and recent data sets [90] might allow a more detailed analysis of the role of different NPIs. Still, it should be noted that, even where similar NPIs were in place, it is not trivial to quantify population adherence or disentangle their effects. Overall, mobility data has been shown to be a good proxy for the strength of lockdown measures and other NPIs [49, 50] and has been increasingly available and used in recent years as an indicator of behavioural changes affecting infection dynamics (e.g. [51–54]).

The COVID-19 pandemic also made clear how understudied some of the NIRVs are. There is extensive evidence from experimental and epidemiological studies on the effects of temperature or humidity on IVs and, despite being more limited, research on NIRVs dynamics also mostly focuses on the effect of weather at the epidemiological level [9, 91–93]. Even for the pandemic period, and as far as we could gather, studies on NIRVs, other than RSV, have been limited to reports on the reductions in the circulation of these viruses immediately after the implementation of the interventions up to January 2022 [80–88].

From our analysis, two different groups of viruses emerged in the pre-pandemic period, both in terms of their epidemiological patterns and their weather susceptibility. One group contains IVA, RSV and hCoVs, with higher apparent weather sensitivity; another groups IVB and hMPV, with lower weather sensitivity. These results are in agreement with other pre-pandemic studies, on a smaller set of viruses, which showed that, in temperate regions, the start of the RSV and IVA epidemics were earlier and closer in time, followed by the start of IVB and hMPV epidemics [9] (with exceptions). The same study also found an association between temperature and RH, and the activity of IVA, IVB and RSV. Another study, using 7 years of data from patients in Edinburgh, Scotland, identified an effect of temperature on the activity of RSV, IVA, IVB and hMPV, being the largest in RSV and the lowest in hMPV [93]. Therefore, the correlation with weather described for IVs, specially IVA, does not generalize so well to different NIRVs (and the temperature/humidity dependence of SARS-CoV2 remains an open debate). Our results showing two groups of viruses in terms of epidemiological dynamics corresponding to the two groups with different climatic sensitivities, raise the question of whether there is a causal relation between the different climatic sensitivities and the different epidemiological patterns, and whether, if the former is true, this will still have any effect after the COVID-19 pandemic. Alternative hypotheses might include the possibility of increased likelihood of infection with a virus from the second group after infection with a virus from the first, but experimental studies on transmission and survival of these different NIRVs, as well as epidemiological studies, including longitudinal cohorts, would be needed to test them.

Overall, weather and mobility affect the different viruses differently. The broader introduction of vaccines against some (IVs, RSV) but not all viruses, together with climate change and the effects that raising temperatures might have on different viruses, might alter their combined dynamics, particularly if there are relevant viral interactions. Our results strongly highlight the importance of further research on different viruses and emphasize that these effects should not be studied in isolation.

In summary, we have shown evidence indicating that mobility patterns played a very relevant role in shaping the dynamics of respiratory viruses during the different phases of the COVID-19 pandemic, prevailing over the effects of weather conditions. Altered viral dynamics with unforeseen surges, as observed in this period, have important consequences for health systems worldwide and, that mobility might offer good predictors of upcoming epidemics argues in favour of public health organizations increasingly including mobility in their models and surveillance systems. Our results reinforce the argument that the seasonal epidemiological dynamics of respiratory viruses are driven by a complex system of interactions between the different factors (weather, human behavior, virus interaction and intrinsic viral factors), which probably led to an equilibrium that was disturbed, and perhaps permanently altered, by the COVID-19 pandemic. Thus, future research will profit from more research on NIRVs and from the implementation of more complex mechanistic models that could tackle these questions.

## Supporting information

**S1 Fig. Total number of tests performed for each virus in Canada between 2016 and 2023 and Pearson correlation coefficients between the two incidence proxies.**
(PDF)

**S2 Fig. Incidence for each virus analized, weather and mobility time series and correlations in the USA.**
(PDF)

**S3 Fig. Incidence for each virus analized, weather and mobility time series and correlations in Canada.**
(PDF)

**S4 Fig. Correlation and PCA analysis for the different weather variables for independent variable selection.** PCA analysis for the US Department of Transportation data set for mobility variable selection.
(PDF)

**S5 Fig. ΔAIC for the temperature and RH models that include one, two, three, four or no week auto-correlation lag variable for the USA and Canada.**
(PDF)

**S6 Fig. Pearson correlation analysis for the different viruses and weather variables lagged 1, 2, or 3 weeks for Canada and the USA, during pre and pandemic periods.**
(PDF)

**S7 Fig. Spearman correlation analysis for the different viruses and weather variables for Canada and the USA, during pre and pandemic periods.**
(PDF)

**S8 Fig. Regression coefficients and pseudo-$R^2$ for the temperature-RH-AC model in the pre-COVID-19 pandemic period from September 2016 to September 2018 in Canada and**

the USA.
(PDF)

**S9 Fig. Incidence of IVA, IVB, hCoVs, HMPV and RSV in Canada and the USA in the pandemic period (starting at April 1st, 2020).**
(PDF)

**S10 Fig. Regression analysis for the temperature and population at home model for the USA in the pre and pandemic periods.**
(PDF)

**S11 Fig. Comparison of the regression coefficients for the temperature and number of trips or population at home models when the pandemic period starts in January 2020 or March 2020.**
(PDF)

**S12 Fig. Regression coefficients for the temperature and number of trips or population at home models with a shorter pandemic period (January 2021-March 2022).** Regression coefficients for the temperature and number of trips model with dummy pandemic variable and interactions.
(PDF)

**S13 Fig. Regression analysis for the temperature and the residential time model for the USA and Canada.**
(PDF)

**S1 Table. Models for the weather analysis for all viruses in the pre-COVID 19 pandemic period in Canada and the USA.**
(PDF)

**S2 Table. Models for the weather analysis for all viruses in the COVID-19 pandemic period in Canada and the USA.**
(PDF)

**S3 Table. Models for the mobility analysis (number of trips) for all viruses in the pre and COVID-19 pandemic period in the USA.**
(PDF)

**S4 Table. Models for the mobility analysis (population at home) for all viruses in the COVID-19 pandemic period in Canada and the USA.**
(PDF)

**S5 Table. Models for the mobility analysis (transit stations) for all viruses in the pre and COVID-19 pandemic period in the USA.**
(PDF)

**S6 Table. Models for the mobility analysis (residential times) for all viruses in the COVID-19 pandemic period in Canada and the USA.**
(PDF)

## Acknowledgments

The authors would like to thank all members of the SPAC lab for comments and critical reading of the manuscript, Eleonora Tulumello, João Gama Oliveira and Pedro Rio for initial

discussions, and Raquel Guiomar, and Ana Paula Rodrigues from the Instituto Nacional de Saúde Doutor Ricado Jorge, for access to preliminary data.

## Author Contributions

**Conceptualization:** Joana Gonçalves-Sá.

**Data curation:** Irma Varela-Lasheras, Sara Mesquita.

**Formal analysis:** Irma Varela-Lasheras, Lilia Perfeito.

**Funding acquisition:** Joana Gonçalves-Sá.

**Methodology:** Lilia Perfeito, Joana Gonçalves-Sá.

**Supervision:** Lilia Perfeito, Joana Gonçalves-Sá.

**Visualization:** Sara Mesquita.

**Writing – original draft:** Irma Varela-Lasheras.

**Writing – review & editing:** Irma Varela-Lasheras, Lilia Perfeito, Sara Mesquita, Joana Gonçalves-Sá.

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
