## [Decision Letter · Decision Letter 0]

11 Aug 2023

PDIG-D-23-00172

The effects of weather and mobility on respiratory viruses dynamics before and during the COVID-19 pandemic

PLOS Digital Health

Dear Dr. Gonçalves-Sá,

Thank you for submitting your manuscript to PLOS Digital Health. After careful consideration, we feel that it has merit but does not fully meet PLOS Digital Health's publication criteria as it currently stands. Therefore, we invite you to submit a revised version of the manuscript that addresses the points raised during the review process.

Please submit your revised manuscript within 60 days Oct 10 2023 11:59PM. If you will need more time than this to complete your revisions, please reply to this message or contact the journal office at digitalhealth@plos.org. Please include the following items when submitting your revised manuscript:

We look forward to receiving your revised manuscript.

Kind regards,

Ryan S McGinnis

Academic Editor

PLOS Digital Health

Journal Requirements:

1. We ask that a manuscript source file is provided at Revision. Please upload your manuscript file as a .doc, .docx, .rtf or .tex.

2. Please provide separate figure files in .tif or .eps format only and remove any figures embedded in your manuscript file. Please also ensure that all files are under our size limit of 10MB.

Additional Editor Comments (if provided):

PLOS Digital Health! Please consider the thoughtful comments from each reviewer in your revisions.

Reviewers' comments:

Reviewer's Responses to Questions

**Comments to the Author**

1. Does this manuscript meet PLOS Digital Health’s publication criteria? Is the manuscript technically sound, and do the data support the conclusions? The manuscript must describe methodologically and ethically rigorous research with conclusions that are appropriately drawn based on the data presented.

Reviewer #1: Yes

Reviewer #2: Yes

Reviewer #3: Partly

2. Has the statistical analysis been performed appropriately and rigorously?

Reviewer #1: Yes

Reviewer #2: Yes

Reviewer #3: I don't know

3. Have the authors made all data underlying the findings in their manuscript fully available (please refer to the Data Availability Statement at the start of the manuscript PDF file)?

Reviewer #1: Yes

Reviewer #2: Yes

Reviewer #3: Yes

4. Is the manuscript presented in an intelligible fashion and written in standard English?

Reviewer #1: Yes

Reviewer #2: Yes

Reviewer #3: No

5. Review Comments to the Author

Reviewer #1: Summary:

This paper presents a multimodal epidemiological analysis leveraging weather, mobility, and time-scale factors on the transmission of respriratory viruses in Canada and the USA from 2016 to 2023. Specifclaly, the team aimed to explore the differences in these factors for viral prevalence and spread pre- and post- the COVID-19 pandemic onset. The authors deploy several computational techniques to demonstrate the differences in effect of factors, between-factor correlations, differentiate transmission trends between - NIRVs versus IVs, and share relevant context for past work in this domain. The authors found that pre-pandemic, weather-related factors had greater statistical effect on transmission over mobility data. Post-pandemic, they found that juxtaposed results – where mobility had a equal or larger effect on viral spread when compared to weather-related factors. 

Strengths:

Overall, I found this work to be easy to read and flow well. I was impressed at the level of support via relevant background studies that was highlighted in this work. I also appreciated that final conclusions were not overstated and that there was a call for more multimodal research to further validate claims. I have just a few minor edits and questions that I feel the authors could/should consider. A few specific strengths of the work is the highilight that increased geographic resolution and coverage was considerably impact the this type of analysis. The emphasis on multimodal complex system analysis to describe the dynamics of respiratory viruses was well supported and I appreciated seeing it highlighted so successfully in the discussion and introduction. 

Weaknesses:

- While it was stated that the start date for the “pandemic period” was selected based on public declaration by the WHO, I wonder if your team considered or tested the pandemic period starting when the virus was first identified in late 2019. As this period from late 2019 to March 2020 did have circulation of SARS-CoV-2, I was surprised it was not included in the pandemic period. 

- Mobility data for “pre-pandemic” only includes data from late 2019 to March 2020, which, as just mentioned, is a period of circulation for SARS-CoV-2, even if the declaration had not been officially given. I found this to be a hole in the work, as mobility can not properly be addressed/assessed for pre-pandemic if this is the only window it is considered within. It is not stated specifically why mobility data was not included or considered for earlier periods. 

- Of the correlation tests completed, was there a specific reason why Pearson was selected over or not in combination with Spearman? I wonder if there was a reason to consider only linear effects and not monotonic associations. 

- Also for your correlations, there are many correlation tests discussed and heatmaps shown, but little if any discussion of the statistical significance of these correlations. Was there statistical significance tested for these Pearson correlation analyses?

- When selecting for autocorrelation (AC) windows, I am curious of why your team only selected 1 week or 2 weeks or no autocorrelation. Did you run any tests for 3 weeks or monthly/4-week patterns? I would suggest adding information for these periods or clarifying why they are not considered. 

- The selection of only one primary mobility variable was not well described. Why was only “number of trips” selected for analysis? This selection of just one factor should be described more thoroughly prior to labeling as “the mobility variable”. Weather factors are heavily discussed and supported, while mobility factors seemed brushed off and not fully supported or explained. Originally, upon introductory description, it seemed like mobility data and its multiple factors would play a larger role in analysis but then it did not. 

- In the conclusion of your regressions section, directly prior to results, you stated that there was previous evidence of the relevance of the RH AC Temperature Model. Please cite this prior work here rather than just saying there is evidence. 

- In the results section, add references to the sentence “as others before us, we found strong correlations between the viruses….”. 

- Results for mobility being such a considerable factor for the COVID-19 pandemic period made the fact that we had no true pre-pandemic mobility data glaring and seems like a major missing key part for this analysis. Though the weather-related factors are well supported, mobility comparisons do not.

- Was there considerations of other NPI factors that could’ve been added to this analysis? The discussion of these factors helped the narrative, but I would have hoped to see at least a statement planning this integration and consideration in future work. The variability in mandates and restrictions, as mentioned, play a huge role in transmission and are a needed component for this analysis to be fully robust. 

- While the relevance for climate change and for broad understanding of disease transmission represent important precipitators to this type of analysis, it would be helpful to see additional support for the “so what” of this factor analysis. Really hammer home why this work is important – because it is!

Again, while these are changes I feel would really help the flow and impact of the work, I overall feel the analyses were sound and that the narrative is well-written and impactful.

Reviewer #2: The manuscript investigates the co-circulation dynamics between seasonal respiratory viruses and the potential drivers of such dynamics, comparing the periods before and during the COVID-19 pandemic. In particular, the role of weather and mobility on the timing of epidemic waves is analyzed through correlation analyses and regression.

Overall results are interesting. However, I believe that the paper requires major improvements before I can recommend its publication. 

The paper reads overall well. However, it is too lengthy and, in many parts, redundant. The description of the methods is in part repeated on the Results. In addition, the results are sometimes discussed and interpreted in the Results section. I believe that the paper would be much more readable if the Results section was more succinct with no repetition of methods and no anticipation of the discussion.

The study is conducted at the country level. Both the USA and Canada are extended countries. In the USA, weather variables vary substantially across different regions. Authors should properly acknowledge this limitation and discuss how this impacts the results. 

Do the Authors consider a lag in the correlation analysis? I believe that given the delays in reporting the correlation analysis could consider a lag of at 1 or 2 weeks.

Although mobility data for the USA are available only for one year during the pre-pandemic period, it seems to me that the range of variation of mobility during the pandemic period is greater than in the pre-pandemic period. This could explain why this quantity was more important in explaining the incidence trend during COVID-19 pandemic. The Authors should discuss this point more in detail.

“it was suggested that the IVB/Yamagata lineage could have become partially extinct, due to the lack of animal reservoirs and other epidemiological characteristics” I find this sentence quite vague. I don’t believe that the lack of animal reservoirs is the main cause of IVB/Yamagata extinction. In addition, “other epidemiological characteristics” is not clear. 

“Furthermore, if some of these viruses are less weather-sensitive than previously believed, and summer peaks are clearly possible for some of them, global changes in weather might alter viral dynamics in ways that are difficult to predict, which is particularly relevant in the context of climate change”: Authors should better elaborate on this point since it is quite important, but at the same time, this sentence does not seem to me well substantiated.

Reviewer #3: The study topic is interesting, but some changes are needed:

1. The study title should clearly define that this study includes data from the US and Canada.

2. The abstract should be more informative and scientifically based. The current version seems to be too simple and plain.

3. It will be better if the text is more professional. So please consider changes e.g., in line 35 where the word "Naturally" seems to be inappropriate. 

4. Data source and management is a little bit unclear. How do the different data collection and reporting methods affect the results? What is the risk of bias and how do the Authors handle it? 

5. Why only temperature and humidity were selected? The methods section should be more clear and easy to follow. Some assumptions may be unclear to the readers. How about the meteorological data for Canada?

6. The Authors used two different approaches and data collection methods for two different countries. There is a serious risk of bias. The Authors may focus only on 1 country and should consider a separate analysis for the US or Canada.

7. Wording like "it is quite possible" or "is reason to believe that" seems inappropriate in the scientific paper.

6. PLOS authors have the option to publish the peer review history of their article (what does this mean?). If published, this will include your full peer review and any attached files.

**Do you want your identity to be public for this peer review?** For information about this choice, including consent withdrawal, please see our Privacy Policy.

Reviewer #1: No

Reviewer #2: No

Reviewer #3: No

---

## [Decision Letter · Decision Letter 1]

7 Nov 2023

The effects of weather and mobility on respiratory viruses dynamics before and during the COVID-19 pandemic in the USA and Canada

PDIG-D-23-00172R1

Dear Prof. Gonçalves-Sá,

We are pleased to inform you that your manuscript 'The effects of weather and mobility on respiratory viruses dynamics before and during the COVID-19 pandemic in the USA and Canada' has been provisionally accepted for publication in PLOS Digital Health.

Best regards,

Ryan S McGinnis

Academic Editor

PLOS Digital Health

Reviewer Comments (if any, and for reference):

Reviewer's Responses to Questions

**Comments to the Author**

1. If the authors have adequately addressed your comments raised in a previous round of review and you feel that this manuscript is now acceptable for publication, you may indicate that here to bypass the “Comments to the Author” section, enter your conflict of interest statement in the “Confidential to Editor” section, and submit your "Accept" recommendation.

Reviewer #1: All comments have been addressed

Reviewer #2: All comments have been addressed

Reviewer #3: All comments have been addressed

2. Does this manuscript meet PLOS Digital Health’s publication criteria? Is the manuscript technically sound, and do the data support the conclusions? The manuscript must describe methodologically and ethically rigorous research with conclusions that are appropriately drawn based on the data presented.

Reviewer #1: Yes

Reviewer #2: (No Response)

Reviewer #3: Yes

3. Has the statistical analysis been performed appropriately and rigorously?

Reviewer #1: Yes

Reviewer #2: (No Response)

Reviewer #3: I don't know

4. Have the authors made all data underlying the findings in their manuscript fully available (please refer to the Data Availability Statement at the start of the manuscript PDF file)?

Reviewer #1: Yes

Reviewer #2: (No Response)

Reviewer #3: Yes

5. Is the manuscript presented in an intelligible fashion and written in standard English?

Reviewer #1: Yes

Reviewer #2: (No Response)

Reviewer #3: Yes

6. Review Comments to the Author

Reviewer #1: All comments have been addressed. Thank you.

Reviewer #2: (No Response)

Reviewer #3: The Authors provided sufficient explanation and the manuscript was significantly improved.

7. PLOS authors have the option to publish the peer review history of their article (what does this mean?). If published, this will include your full peer review and any attached files.

**Do you want your identity to be public for this peer review?** For information about this choice, including consent withdrawal, please see our Privacy Policy.

Reviewer #1: No

Reviewer #2: No

Reviewer #3: No
